# Torsion Test vs. Other Methods to Obtain the Shear Strength of Elastic-Plastic Adhesives



Monica Ferraris [1,2], Milena Salvo [1,2], Valentina Casalegno [1,2], Stefano De La Pierre [1,2], Luca Goglio [2,3] and Alessandro Benelli [1,2,*]

1. Department of Applied Science and Technology, Politecnico di Torino, I-10129 Torino, Italy; monica.ferraris@polito.it (M.F.); milena.salvo@polito.it (M.S.); valentina.casalegno@polito.it (V.C.); stefano.delapierre@polito.it (S.D.L.P.)
2. J-Tech@PoliTO—Advanced Joining Technologies@PoliTO, I-10129 Torino, Italy; luca.goglio@polito.it
3. Department of Mechanical and Aerospace Engineering, Politecnico di Torino, I-10129 Torino, Italy
* Correspondence: alessandro.benelli@polito.it; Tel.: +39-(388)-656-0891

**Abstract:** Nowadays adhesive joints are more and more used; therefore, a precise and reliable shear strength measurement of these joints is necessary to design and predict a final components' performance. This work aimed to assess the shear strength value of adhesively joined ceramics (SiC, Si3N4) and steel in the case of an elasto-plastic (ductile) joining material (Loctite EA 9321 AERO) by an experimental campaign and associated analytical modelling. The joined samples were tested using a single lap offset test in compression (SLO), an asymmetrical 4-point bending test (A4PB, ASTM C1469), and by torsion on fully joined hourglass shaped samples (THG). A simple model based on the elastic-plastic response in shear was proposed to fit the torque-rotation curve measured in the torsion tests. The results showed that, with the adopted test methods and conditions, and by using the model, consistent values of shear strength could be obtained by torsion tests.

**Keywords:** joining; adhesive; torsion; mechanical tests





## 1. Introduction

Assessing the shear properties, in particular the strength, of an adhesive is a goal of great interest to achieve, as witnessed by several contributions in the past and recent literature [1–3]. Indeed, before other more sophisticated parameters, such as the fracture toughness, the shear strength is the first design value of practical use, especially in an industrial scenario. This static parameter is very important because it can be taken as a starting point for more in-depth analyses, such as the cyclic response [4].

Additional problems arise in the case of brittle (or nearly brittle) adherends such as ceramics, due to the difficulties in manufacturing the specimens and applying the test load. For this class of materials, a specific standard is available [5] based on the 4-point asymmetric bending (A4PB in the following) test. Its shortcomings, thoroughly discussed in [6], are briefly summarized here: the samples for A4PB tests need precise alignment during preparation and testing (which can be difficult in the case of miniaturized samples); they should not be prepared one by one; A4PB is not usable if the bond strength is greater than 50% of the adherend bending strength; and "U" or "V" notches are required in the joined region if the bond strength is greater than 25% of the adherend bending strength.

Obtaining a reliable value of shear strength for joined samples, not affected by other stress components and stress concentrations present in the commonly used tests, is a well-known problem: in this view, simple lap shear tests are not suitable, as recognized also by the disclaimers present in some standards, e.g., ASTM D905, ASTM D1002 [7,8]. Therefore, results from lap tests should, in general, be used only for comparative purposes and not to evaluate the shear strength for design purposes.

A particular type of lap shear test is the single lap offset (SLO in the following) test in compression, a modification of the ASTM D905 (designed for wood substrates) in which, together with thick and stiff substrates, ad hoc fixtures are adopted to keep these substrates as parallel as possible during the test to minimize peel stress due to rotation and/or bending of the substrates [6].

Torsion represents a suitable alternative to other methods to provide reliable shear strength values. Recently, an hourglass shape of the specimen (THG in the following) was proposed, which allowed one to obtain a bond of a reduced circular section. Advantages of this choice, reported in references [9–11], are mainly represented by the pure (although not uniform) shear stress state for the joined section under torsion, and by the possibility to obtain failure in the bond in case of a high-strength joining material.

Another popular configuration for adhesive testing is given by the Arcan fixture [12,13], in which the joint specimen, formed by two (relatively thick) plates bonded by an adhesive layer, can be loaded under different adjustable directions, which are variable between pure normal (tension or even compression) and pure shear, with intermediate "blended" cases. Special modifications have been designed [14] to minimize edge effects in the bondline ends. However, in the scenario of the present work, the Arcan scheme would not behave differently from a lap shear scheme, therefore it was not adopted.

A previous paper from our group [10] considered the features of the torsion test in the case of a brittle adhesive and THG specimen. In particular the advantage represented by the lack of stress singularity at the adhesive–adherend interface, and a moderate notch effect ($K_t$ = 1.2–1.3), was due to the non-constancy of the cross section.

Most of adhesives' technical datasheets just provide lap shear strength values for joined components, often without specifying the way it is measured and if the adhesive has plastic or brittle behavior.

This paper tries to fill this gap by providing a procedure for evaluating the "true" shear strength for adhesively joined components.

In particular, the attention was focused on the case of a ductile adhesive, which added the need for including its plastic response in the study.

The experimental activity involved Hysol EA9321, taken as representative of this class of product, and three different materials as adherends, namely two ceramics and a steel.

The main goal was to compare the response of the three above-mentioned test schemes—A4PB, SLO, THG—and investigate the consistency of what was obtained.

It was assumed that the adhesive behaviour was ductile and that its stress–strain response under shear could be described by an elastic-perfectly plastic law.

Therefore, the adhesive property that was sought was the elastic limit stress in shear, $\tau_{el}$.

## 2. Materials and Methods

The adhesive used in this work was Hysol® EA9321 [15] (Henkel Corporation, Bay Point, CA, USA), a two-component thixotropic bicomponent adhesive, which was cured at room temperature according to the supplier specifications. Its nominal lap shear strength measured according to ASTM D1002 2024-T3 clad aluminum was 27.6 MPa at room temperature.

The three materials chosen for the adherends were:

- Si3N4, silicon nitride, produced by FCT Ingenieurkeramik GmbH (Germany), a polycrystalline β-Si3N4 obtained by gas pressure sintering using 3–10 wt% of sintering additives; its chemical composition and properties can be found in reference [16].
- SiC, silicon carbide Boostec® SiC (former SiC100®) produced by Mersen (France), a polycrystalline α-SiC (>98.5 wt% SiC) obtained by pressureless sintering; the chemical composition and properties of Boostec® SiC can be found in reference [17].
- Crofer® 22 APU steel produced by ThyssenKrupp VDM GmbH (Germany), a high-temperature ferritic stainless steel especially developed for application in solid oxide

fuel cells (SOFC); its chemical composition and properties can be found in reference [18].

All materials were cut as shown in Figure 1a–c to obtain parts to be joined for SLO (a), A4PB (b), and torsion tests (c); the dimensions of the joined samples were reported.

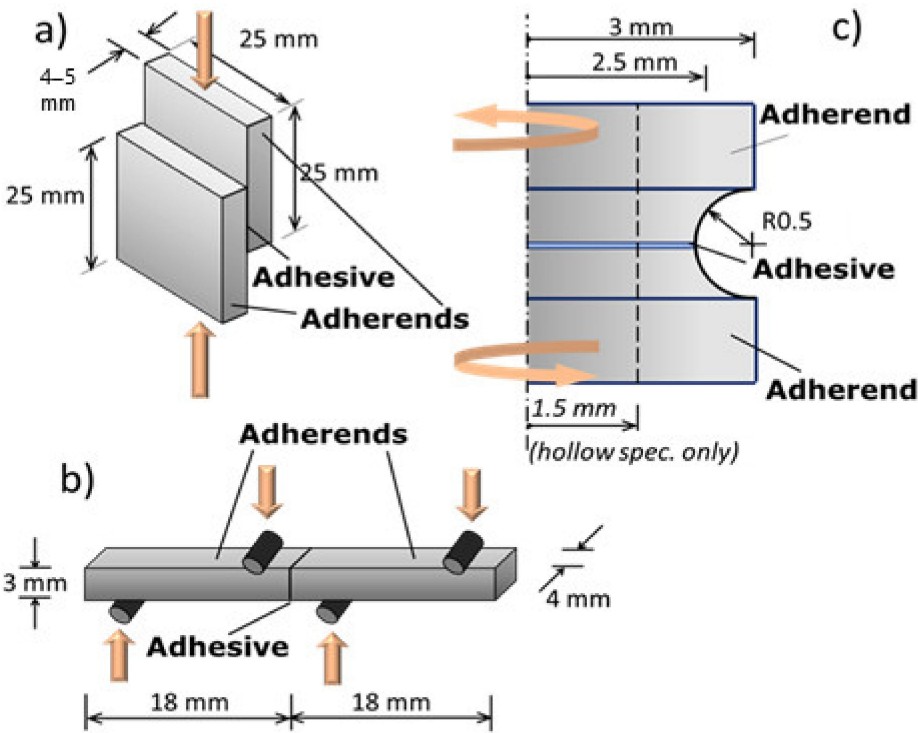

**Figure 1.** Schematic representation of the three mechanical tests used for the comparison of shear strength measurements on EA9321 joined samples: (**a**) single lap offset (SLO); (**b**) asymmetric 4-point bending (A4PB) ASTM C1469; (**c**); torsion on hourglass-shaped samples with full joined (THG) or an annular joined area (TDHG). All sizes are in mm.

The SiC and steel samples were grinded, then polished to 1 μm using diamond paste, and sonicated in acetone. The Si3N4 samples were surface engineered as described in reference [19] (unfortunately the same could not be obtained for SiC, as discussed in the reference). All samples were then bonded according to the instructions reported in the EA9321 data sheet [15].

*2.1. Test Procedures*

The shear strength of all joined samples was measured at room temperature on at least 5 nominally identical samples with each of the above-mentioned techniques: SLO, A4PB, and THG.

All tests were carried out by using a universal testing machine SINTEC D/10 with the suitable different fixtures and crosshead speed for each of the three tests. In particular, in the case of torsion, a specific test rig [9] was used, to transform the tensile loading applied by the testing machine into torque. Figure 2 shows the torsion apparatus used to perform torsional tests. It was possible to identify the load transmission chain (1), the sample grips (2), and the 2kN load cell (3). The rotational speed used in the tests was 0.65 degree/min, which corresponded to a controlled crosshead speed of 0.5mm/min.

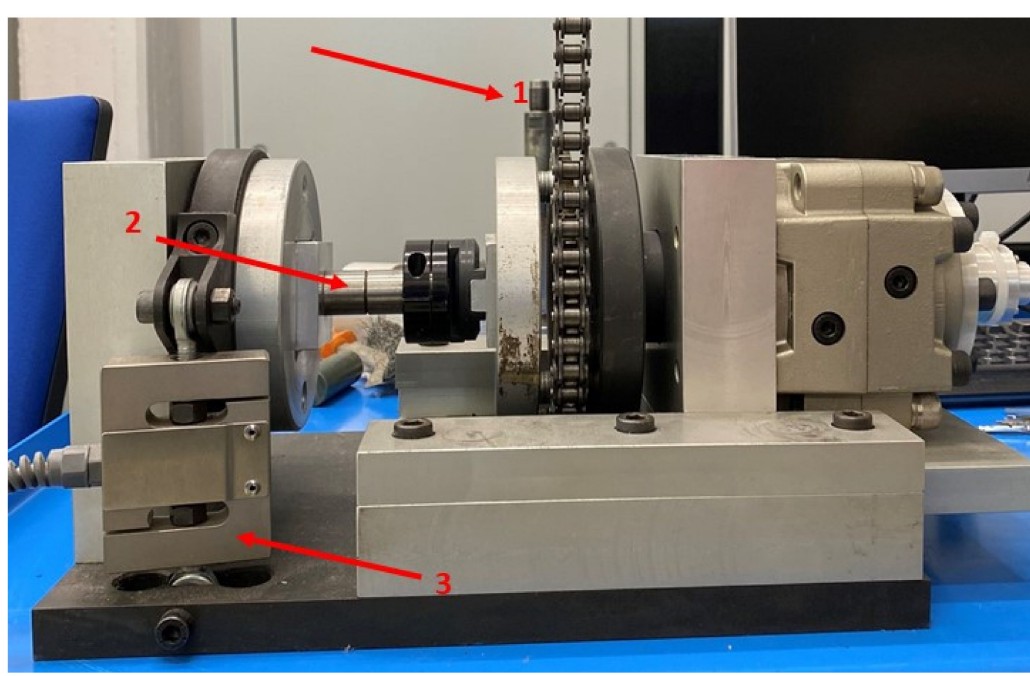

**Figure 2.** Torsion test apparatus.

All samples failed in the joined region, in the cohesive or interfacial mode, as reported in Table 1.

**Table 1.** Results after mechanical tests on Hysol EA9321 joined samples. Torsional (THG) shear strength results obtained by analytical modelling (in grey, in parentheses, are the results obtained by using the maximum of the torsion curve, for discussion purposes only).

| | Test Type | | |
|---|---|---|---|
| **Adherends** | **SLO (MPa)** | **A4BP (MPa)** | **THG (MPa)** |
| Si3N4 (*) | 42.2 ± 1.3 | 41.6 ± 8.0 | 44.5 ± 4.4 (62.6 ± 5.0) |
| Sic | 41.6 ± 0.9 [20] | 41.5 ± 5.0 | 46.5 ± 3.7 (62.0 ± 4.0) |
| Steel | 40.8 ± 4.6 | 39.0 ± 3.0 | 52.2 ± 4.6 (67.0 ± 6.0) |

(*) engineered surface cohesive failures [19]; all other failures are interfacial.

### 2.2. SLO—Single Lap Offset

The SLO in compression test (Figure 1a) is adapted from the standard ASTM D905-08 and also described in [21]. The fixture was designed to minimize the substrates' bending and rotation, thus obtaining a parallel relative displacement between them, which generated pure shear (Figure 3). The much higher stiffness of adherend materials (*E* moduli: Si4N4 320 GPa, SiC 400 GPa, steel 206 GPa) with respect to the adhesive (*E* modulus 4 GPa) contributed to approach the condition of deformable adhesive between stiff adherends, with a consequent, nearly uniform, stress along the bondline. In addition, as the adhesive was ductile, the unavoidable stress peaks at the interface ends could also be compensated for by plasticity when the joint yielded.

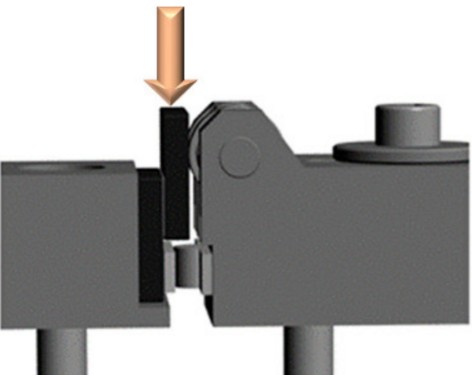

**Figure 3.** Fixtures to avoid bending and rotation of the adherends in the SLO test.

In this perspective, evaluating the shear strength $\tau_{el}$ as a load-to-area ratio gave not only a conventional value but also an actual material property of the adhesive.

Tests with these SLO fixtures were performed at a controlled displacement of 0.5 mm/min.

### 2.3. A4PB—Asymmetric 4-Point Bending

Preliminarily, a (symmetric) 4-point bending test was performed according to ASTM C1161 [22] to assess the flexural strength of the joint: all values were lower than 25% flexural strength of the substrates (Si3N4, SiC, and steel); thus a specimen geometry without V-notches for the A4PB could be used.

The A4PB was performed according to ASTM C1469 [4] (Figure 1b). This testing scheme produced a condition of shear force only (nil bending moment) in the test section containing the bond. The stress in a section subject to shear force was notoriously non-uniform in the elastic case; however, taking advantage of this kind of test method for the ductile behaviour of the adhesive that levelled the stress distribution, the shear strength $\tau_{el}$ was calculated by using Equation (1) [4]:

$$\tau_{el} = \frac{P_{max}(S_o - S_i)}{A(S_o + S_i)} \tag{1}$$

where $P_{max}$ was the ultimate load, $A$ was the bonded area, $S_o$ and $S_i$ were, respectively, the outer and inner span between the rollers in the scheme of Figure 1b.

Asymmetric 4-point bending tests were performed at a controlled displacement of 0.3 mm/min.

### 2.4. THG—Torsion Hourglass

Torsion of a circular section creates a state of pure shear stress, as stated previously, with its peak value in the periphery. Once the maximum shear stress was achieved, the yield limit $\tau_{el}$ and the applied torsional moment was increased and a plastic annular zone was formed in the outer part of the cross section from radius $R^*$ to radius $R_e$, whilst a circular zone (or annular, if the section was hollow with inner radius $R_i$) in the central part of the cross section remained elastic. Figure 4 depicts the situation.

The total torsional moment applied in such a condition could be described as the sum of two contributions: an elastic term $M_{el}$ corresponding to the elastic shear stress distribution in the inner part of the cross section, and a plastic term $M_{pl}$ corresponding to the constant shear stress in the outer part of the cross section:

$$M_{tot} = M_{el} + M_{pl} = \tau_{el}\frac{\pi\left(R^{*4} - R_i^4\right)}{2R^*} + \tau_{el}\frac{2\pi}{3}\left(R_e^3 - R^{*3}\right) \tag{2}$$

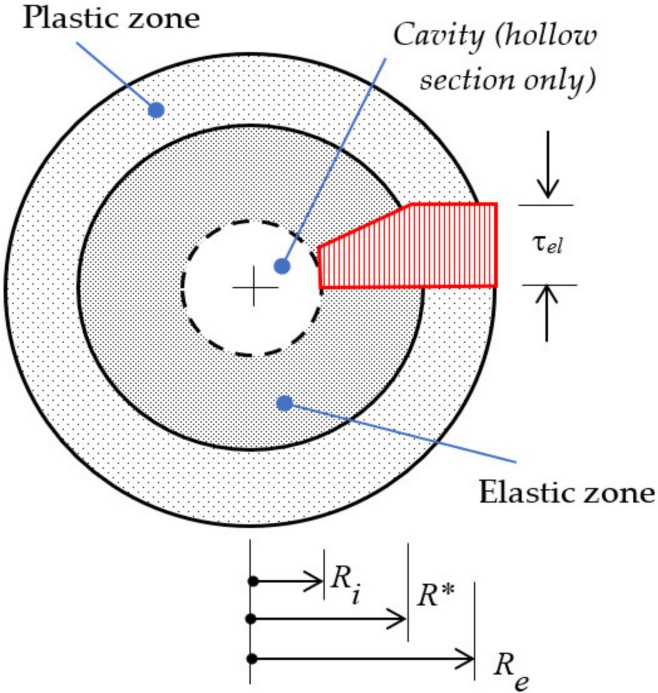

**Figure 4.** Elastic-plastic model of the section under torsion.

From Equation (2) it could be easily noticed that when $R^* = R_e$ plasticity occurred at the outer edge and yielding started, whilst when $R^* = R_i$ the ultimate value of the torsional moment was reached, and the section was completely yielded.

The twist gradient $\theta'$ is governed by the elastic zone only, according to the known equation:

$$\theta' = \frac{M_{el}}{GJ^*} = \frac{\tau_{el}}{GR^*} \tag{3}$$

In the case of a non-hollow section, $R_i = 0$, the ratio of the first yield moment $M_y$ to the ultimate moment $M_{max}$ read

$$\frac{M_y}{M_{max}} = \frac{\tau_{el}\pi R_e^4/2R_e}{\tau_{el}2\pi R_e^3/3} = \frac{3}{4} \tag{4}$$

which corresponded to the classical relationship by Nadai (1931) recalled in [2].

### 3. Results and Discussion

The only relevant results recorded from the SLO and A4PB tests were the ultimate load values, that simply divided by the area or applied Equation (1), respectively, giving the shear strength $\tau_{el}$.

### 3.1. Torsion Tests

More interesting were the results of the THG tests reported in terms of torsional moment vs. rotation angle diagrams. Figure 5a–c displays the experimental curves and the corresponding model counterparts.

### 3.2. Elasto-Plastic Torsion Test Model

The producer reported a tensile lap shear strength (measured according to ASTM D1002 [7]) of 27 MPa on Hysol-joined Al samples tested at room temperature (25 °C) after curing. Reported bulk properties for casted and cured Hysol samples were as follows: a tensile strength of 49 MPa with an elongation at break of 6% and an ultimate compressive

strength of 116 MPa, with a yield in compression of 64 MPa (measured according to ASTM D695), indicating a certain elasto-plastic behavior [15].

One of the purposes of this paper was to propose the correct use of results obtained by torsion tests on joined components. In the case of the elastic-plastic behaviour of the joining material, the shear strength measured by torsion could not be calculated by treating the mechanical behaviour of the joint, e.g., a fully elastic response considering just the maximum torque of the experimental curve. Calculating the shear strength with the elastic equation obviously led to an overestimation of it, as was evident from the THG values in parentheses in Table 1. Moreover, the strength (incorrectly) obtained in this way was size dependent, as the result was also influenced by the diameter of the specimen. These features were observed and reported in [23] for a different adhesive (AV119) and briefly discussed therein as the effect of plastic behaviour for that adhesive.

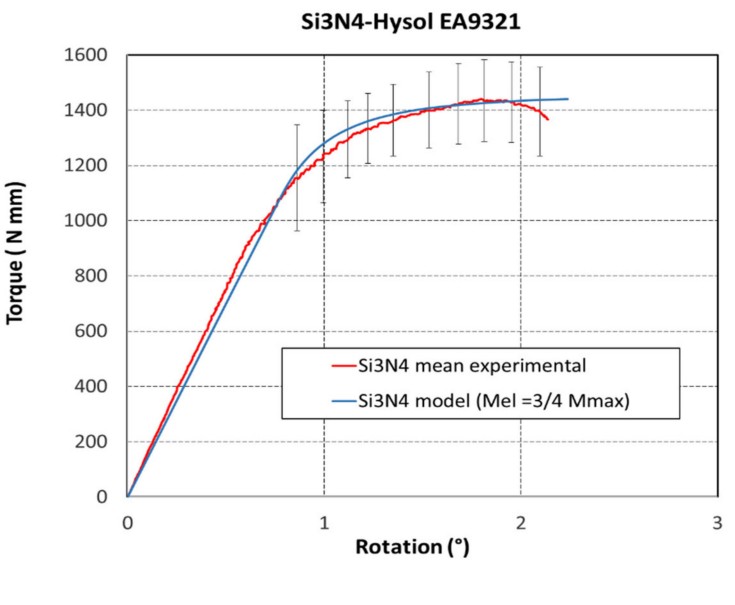

(**a**)

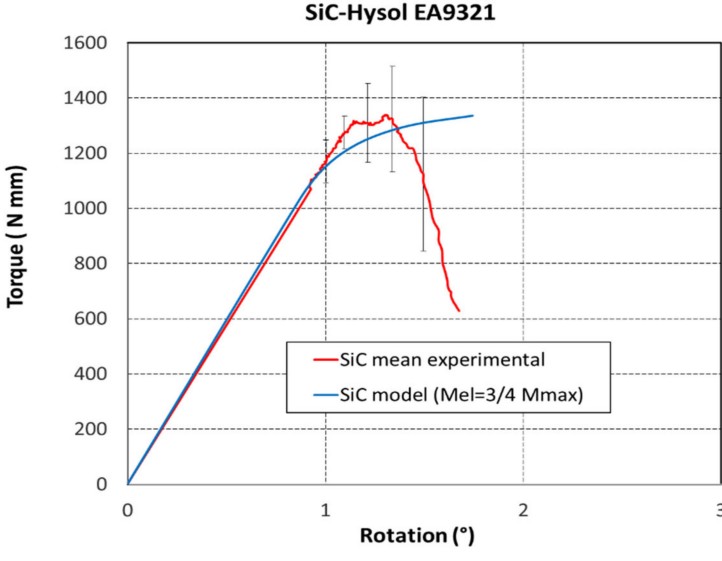

(**b**)

**Figure 5.** *Cont.*

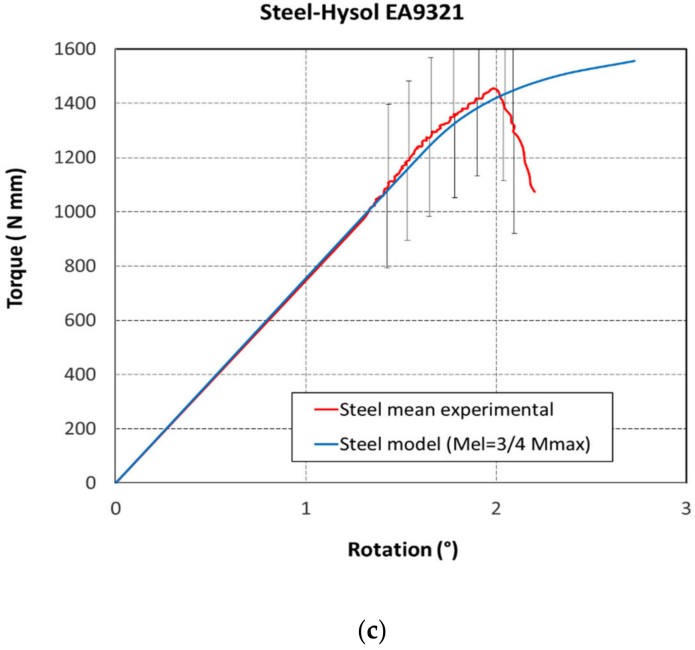

(**c**)

**Figure 5.** Mean experimental and model curves for the THG samples under torsion: cases of adherends in Si3N4 (**a**), SiC (**b**), and steel (**c**). Vertical bars stand for ±one standard deviations. Cohesive failure of the surface-structured Si3N4 (**a**) showed an evident plastic behaviour in the experimental torsion curve, while interfacial failure of SiC (**b**) and steel (**c**) corresponded to a limited plastic region.

Reaching elastic limit did not imply immediate failure of the elastic-plastic joining material: in this perspective, failure occurred when the whole section had yielded, provided that the joining material was ductile enough not to fracture up to such a condition.

To obtain experimentally reliable values for the shear strength through a torsion test, avoiding the use of elastic-plastic relationships (2)–(4), ideally the spread of plasticity should be prevented. This could be performed experimentally by creating a hole in the centre of the hourglass specimen, as shown in Figure 6.

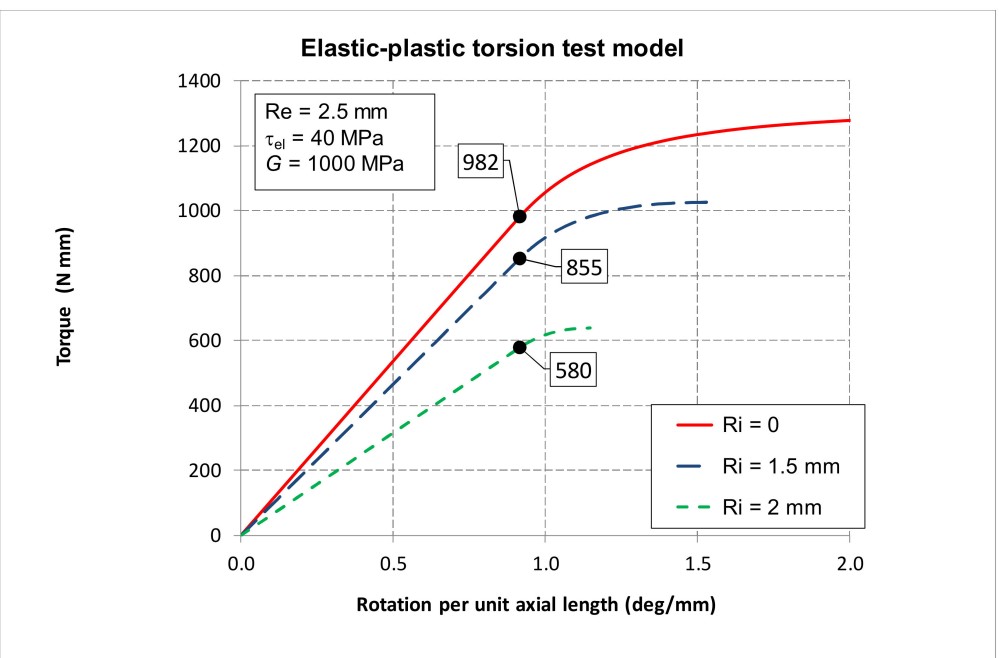

**Figure 6.** Modelling the effect of the internal radius on torsion tests results.

Application of the model clearly showed that in the case of a very thin joined annulus ($R_i \rightarrow R_e$) the plastic contribution was minimal and the shear strength $\tau_{el}$ could be obtained similarly to under elastic behaviour. Thus, the result was also size independent and was not influenced by the diameter of the specimen.

So, in a purely experimental approach, the true shear strength (size independent) could be obtained by using a ring shaped TDHG and gradually reducing the radial thickness of the joined annulus, so reducing the plastic contribution.

However, experimental results on TDHG samples with a drilled hole of 3 mm gave values higher than those expected by modelling, probably due to the experimental difficulty in preparing such TDHG samples, where some adhesive may spread outside the joined area due to the small size of the specimen and its central drilled hole. Due to the practical impossibility of joining TDHG samples with a drilled hole of 4 mm ($R_i$ = 2, Figure 5), i.e., an annulus 0.5 mm thick, the expected size independence was not obtained with this hourglass geometry and results were not reported here. Further experimental activity is ongoing.

In order to avoid the time consuming (and sometimes experimentally impossible) activity of gradually reducing the radial thickness of the joined ring, the model presented above showed that the elastic limit could be obtained in fair agreement with the experimental torque/rotation curves (Figure 5a–c) and with A4PB results, which provided the correct shear strength for all the joined samples (Table 1).

It is worth noting that the cohesive failure of the surface structured Si3N4 showed an evident plastic behaviour in the experimental torsion curve, while interfacial failures of SiC and steel joined samples exhibited a limited plastic region in the experimental curves.

This strikingly different shape of the experimental torsion curves in the case of interfacial or cohesive failure of the same adhesive could be explained as follows. In the case of cohesive failure, the plastic deformation proceeded from the edge (outside diameter) to the centre of the hourglass. Instead, in the case of interfacial failure, this progressive extension of the plastic zone could not take place and the experimental torsion curve dropped earlier. This also led to an overestimated value of the shear strength, as can be noticed from Table 1, especially in case of steel; the value 52.2 MPa obtained from THG was out of the range in respect to the other results. Actually, the elastic-plastic behaviour assumed by the model implied a rounded upper part of the curve, which did not match with the sudden drop that was experimentally observed in the case of interfacial failures. Thus, trying to replicate this with the model, at least the first part of the experimental post-yield curve (i.e., immediately after yield and before the drop), led to the adoption of a yield stress value higher than the actual one, to "lift" the model curve enough.

### 3.3. Comparison of Different Tests

It was also remarkable to see that the SLO in compression used in this work (Figure 2, designed to constrain and avoid plane bending) gave similar results as those obtained by both A4PB and correctly used torsion.

This similarity could be explained because in the present work a peculiar version of SLO under compression (Figure 2) [20] (derived from ASTM D905–08 [7]) was adopted, which did not suffer from the common problems affecting a lap joint under compression, such as non uniform stress–strain distribution with peaks in the ends of the overlap, and the presence of peel stress due to the possible non-alignment of the adherends and their bending deformability. The two joined adherends were relatively thick and short, thus their bending was minimized. Moreover, the fixture kept the specimen aligned during the test by means of a roller supporting its back face; therefore, a nearly uniform relative sliding of the adherends was obtained during this test.

This condition was similar and even more favourable to the case of the thick adherend shear test ASTM D5656 in tension, which aimed to achieve a condition of nearly pure shear [24].

Thus, by testing an elastic-plastic adhesive able to "smooth" the stresses, and by using a carefully designed SLO, as the one described in this paper, the ultimate load was obtained

under a nearly uniform shear stress distribution that was equal to the elastic limit of the adhesive, i.e., its shear strength.

In the case of the asymmetric 4-point bending (A4PB), the elastic shear stress distribution in the joint section was parabolic, exhibiting at its mid-height the peak value equal to 1.5 of its mean value. With an elastic-plastic adhesive, plasticity progressively extended under an increasing load from mid-height to the joined section edges and the ultimate load was, again, attained under a uniform shear stress distribution, equal to the elastic limit of the adhesive, i.e., the shear strength.

Thus, in the final situation, being similar for both tests (this SLO and A4PB), it is not surprising that the results were nearly equal.

In general terms, it is usually found that interfacial failures lead to lower strength values than cohesive ones, as the potential of the adhesive is not completely exploited. However, in this case the strength of the interface was most likely close to the intrinsic strength of the adhesive, leading to nearly equivalent results in terms of shear strength.

As already discussed above, it was remarkable that the experimental torsion curves, in the case of interfacial or cohesive failure of the adhesive, showed a very different behaviour, which could be explained in terms of partial or full exploitation of the adhesive plasticity, respectively.

Most of all, the plastic failure must start and propagate in the joined region, in a cohesive mode. Results from failures that occurred in interfacial (or mixed cohesive-interfacial as well) were suspicious, as in such a case the stress distribution was not homogenized by plasticity. Samples with failure starting and/or propagating in the substrates should not be considered when calculating the shear strength; in these cases, results might be defined "torsion resistance" of the joined sample, and the joint strength might be higher than the substrate strength.

Table 1 summarizes results obtained by SLO, A4B, and torsion (THG only) on Hysol-joined samples: all failures were interfacial, except those for surface-engineered Si3N4, which were cohesive, as discussed in [19].

The results in Table 1 were calculated considering the maximum of each test curve for SLO and A4PB; in the case of torsion (THG), the reported shear strength results were obtained by using Equations (2)–(4) of the model.

### 3.4. Torsion Test Protocol with Brittle Adhesives

If the adhesive was brittle (purely elastic), once the elastic limit was reached and immediate (catastrophic) failure occurred, and the joint failure was due to the maximum principal stress in the 45° skew plane. If the brittle joining material was weaker than the adherends, the failure was "confined" in the bond region; however, if the adhesive was as strong as, or stronger than, the adherends, a cone-shaped failure occurred in the joined substrate [25]. A circular full joined sample could be used (hourglass, rod, tube, ... ). In the case of an hourglass shape, proper finite element modelling could be required to assess the shear stress distribution in the bond section and to calculate its stress concentration factor $K_t$ (in the case of the THG, $1.21 \leq K_t \leq 1.29$, found in [9]). It is worth noting that with this geometry no singular behaviour at the substrate/joining material interface occurred under torsion [26]. Thus, the shear strength of the joined sample could be calculated as:

$$\tau = K_t \frac{M}{JR} \tag{5}$$

where $K_t$ was the stress concentration factor, $M$ the applied torsional moment, $J$ the polar moment of inertia, and $R$ the external radius.

A ring-shaped bond in this case did not offer any advantage; on the contrary, it added possible manufacturing defects that could cause premature failure.

### 3.5. Torsion Test Protocol with Ductile Adhesive

If the adhesive was ductile (elastic-plastic), in a fully experimental approach a ring (annular) shaped joined sample should be used (tube or drilled hourglass) [8]. Tentative ring width has to be tested and gradually decreased in the annular section width until a constant strength value was obtained. The shear strength of the joined sample in this case could be computed as in Equation (5). Unfortunately, in practice, fabrication of these sample joints was often challenging and the effect of the plastic response might be not completely eliminated. Therefore, in such a scenario the model discussed in this paper was of help in predicting the correct shear strength in the case of elastic-plastic joining material. Obviously, the results should be compared to those obtained from other test methods (SLO, A4PB, . . . ), when available.

It must be underlined that the ductile protocol could also be suitable in the case of brittle joining materials tested at high temperature (e.g., glasses, glass ceramics, or polymers as joining materials, when tested above their glass transition temperature, [27]), as their behaviour couldn't be regarded as ductile.

In the case of non-cohesive failures, measured values gave, nonetheless, an indication on joint strength for that particular adherend–adhesive pair and surface preparation. However, the obtained values could be lower than those obtained from cohesive failure typically achievable when the optimal surface treatment was applied.

## 4. Conclusions

On the basis of an experimental campaign and an analytical model, this work tried to shed some light on the assessment of the shear strength in the case of a ductile (elastic-plastic) joining material. Logically, different testing techniques, if correctly applied, should give the same result, with the same failure mode.

The starting point accounted for the stress distribution, under an elastic regime, for the different specimens: (i) the torsion hourglass specimen exhibited a slightly non-linear distribution that increased from its centre to the periphery; (ii) the 4-point asymmetric specimen exhibited a parabolic distribution, which peaked at mid-height; (iii) the single lap offset specimen exhibited a nearly constant stress distribution (apart from peaks in the ends of the overlap) in the version adopted within this work, which created a nearly uniform longitudinal relative displacement of the adherends, thanks to their stiffness and to the particular fixtures used.

When the joining material yielded, under a perfectly plastic regime the stress distributions in the related zones became approximately constant, making it easy to assess the shear strength. The same applied also to the single lap offset specimen, provided that the joining material was ductile enough to smooth local stress and concentrations in the overlap ends (as it occurred with the adhesive tested in this work). If this was not the case (e.g., different fixtures, brittle joining materials, etc.), the single lap offset test, or other lap tests, should be used for comparative purposes only and not for assessing the shear strength of the joining material.

Thus, if the torsion hourglass test is adopted, the experimental strategy to deal with the effect of plasticity would be to use hollow specimens (TDHG) with increasing inner diameters until the obtained strength (simply evaluated as in the elastic regime) converged to a constant value. However, difficulties can arise from the practical viewpoint, as bonding the specimen on a thin ring-shaped area is challenging. Therefore, the simple model presented in this work can be used to evaluate the shear strength by fitting the measured moment–rotation curve obtained from a full specimen (THG) with the analytical one. It can be remarked that in this way the shear modulus can also be obtained by fitting the initial linear part of the curve.

**Author Contributions:** Validation, M.S., V.C. and S.D.L.P.; formal analysis, L.G. and A.B.; data curation, L.G. and A.B.; writing—original draft preparation, M.F. and L.G.; writing—review and editing, A.B.; supervision, M.F. and M.S.; funding acquisition, M.F. All authors have read and agreed to the published version of the manuscript.

**Funding:** The research leading to these results received funding from the European Community 7th Framework Programme (FP7.2007–2013, call identifier FoF.NMP.2013-10) under grant agreement n° 609188, within the project "Advanced MAnufacturing routes for metal/COMposite components for aerospace" (ADMACOM, https://cordis.europa.eu/project/id/609188/it (accessed on 15 March 2022)).

**Conflicts of Interest:** The authors declare that they have no known competing financial interests or personal relationships that could have appeared to influence the work reported in this paper.

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
