# Peer review of "Torsion Test vs. Other Methods to Obtain the Shear Strength of Elastic-Plastic Adhesives"

_applsci, doi:10.3390/app12073284_

Round 1

Reviewer 1 Report

The authors submitted a good good, although some issues require attention. Please see my enclosed PDF report with comments and issue that require improvement.

Reviewer 2 Report

The paper is interesting and fits with the scope of journal. Minor comments are reported below.

1)The reviewer suggests to improve the quality of abstract.

2) The reviewer suggests to improve the description of the experimetal procedure. The test where performed in displacement or force control? At which rate? Which testing machine? Etc...

3) The reviewer suggests to introduce some photos of experimental tests.

4)The reviewer suggests to expand the characterization of adhesive joints in the introduction. See below

  • Modelling of a GFRP adhesive connection by an imperfect soft interface model with initial damage. Composite Structures 239 (2020) 112034
  •  Cyclic behaviour modelling of GFRP adhesive connections by an imperfect soft interface model with damage evolution. Composite Structures 279 (2022) 114741

Round 2

Reviewer 1 Report

I am happy with the modifications